# Development and Validation of a Deep Learning Model for Histopathological Slide Analysis in Lung Cancer Diagnosis

**DOI:** 10.3390/cancers16081506

**Published:** 2024-04-15

**Authors:** Alhassan Ali Ahmed, Muhammad Fawi, Agnieszka Brychcy, Mohamed Abouzid, Martin Witt, Elżbieta Kaczmarek

**Affiliations:** 1Department of Bioinformatics and Computational Biology, Poznan University of Medical Sciences, 61-806 Poznan, Poland; elka@ump.edu.pl; 2Doctoral School, Poznan University of Medical Sciences, 61-806 Poznan, Poland; mmahmoud@ump.edu.pl; 3Spider Silk Security DMCC, Dubai 282945, United Arab Emirates; 4Department of Clinical Patomorphology, Heliodor Swiecicki Clinical Hospital of the Poznan University of Medical Sciences, 61-806 Poznan, Poland; 5Department of Physical Pharmacy and Pharmacokinetics, Poznan University of Medical Sciences, 60-806 Poznan, Poland; 6Department of Anatomy, Poznan University of Medical Sciences, 60-806 Poznan, Poland; martin.witt@mailbox.tu-dresden.de; 7Department of Anatomy, Technische Universität Dresden, 01307 Dresden, Germany

**Keywords:** lung cancer, deep learning, machine learning, histopathology, convolutional neural networks, medical diagnosis

## Abstract

**Simple Summary:**

A prolonged diagnosis of lung cancer can hinder effective treatment processes for cancer patients. Artificial intelligence-based models significantly impact the healthcare system; deep-learning algorithms in the diagnostic process can save time and money and provide high-accuracy results that accelerate and improve the treatment journey.

**Abstract:**

Lung cancer is the leading cause of cancer-related deaths worldwide. Two of the crucial factors contributing to these fatalities are delayed diagnosis and suboptimal prognosis. The rapid advancement of deep learning (DL) approaches provides a significant opportunity for medical imaging techniques to play a pivotal role in the early detection of lung tumors and subsequent monitoring during treatment. This study presents a DL-based model for efficient lung cancer detection using whole-slide images. Our methodology combines convolutional neural networks (CNNs) and separable CNNs with residual blocks, thereby improving classification performance. Our model improves accuracy (96% to 98%) and robustness in distinguishing between cancerous and non-cancerous lung cell images in less than 10 s. Moreover, the model’s overall performance surpassed that of active pathologists, with an accuracy of 100% vs. 79%. There was a significant linear correlation between pathologists’ accuracy and years of experience (r Pearson = 0.71, 95% CI 0.14 to 0.93, *p* = 0.022). We conclude that this model enhances the accuracy of cancer detection and can be used to train junior pathologists.

## 1. Introduction

Lung cancer remains a significant global health burden, with its incidence and mortality rates steadily rising. According to recent studies, lung cancer is almost three times more likely to kill men than prostate cancer and three times more likely to kill women than breast cancer [1]. The World Cancer Research Organization reported 2,206,771 incidences and 1,796,144 deaths in 2020 [2]. Accurate and timely diagnosis is crucial in determining appropriate treatment strategies and improving patient outcomes. However, the complexity and subjectivity associated with traditional diagnostic methods have led to the exploration of advanced technologies, such as machine learning, to enhance diagnostic accuracy, efficiency, and objectivity.

Nowadays, artificial intelligence (AI) with its subtypes, machine learning (ML) and deep learning (DL) models, particularly those using deep learning approaches, have emerged as powerful tools in histopathology for the detection and diagnosis of lung cancer cells [3]; these tools can provide demand assistance in a patient’s treatment journey, starting from the diagnosis process to the selection of the treatment protocol. In addition to understanding the technical aspects, it is essential to explore the pathologists’ perspectives and experiences of AI and ML in medical diagnosis to ensure their full collaboration with the AI applications and understand the required attributes in the model to perform the tasks functionally [4].

Previously released models show a possible risk of overfitting the training data, resulting in brittle, degraded performance in specific settings. Moreover, it is common for ML-based models to have a tradeoff between accuracy and intelligibility. The highly accurate models are usually not intelligible, while more intelligible models usually provide lower accuracy [5]. Therefore, our study aims to use deep learning algorithms, including convolutional neural networks (CNNs), to develop an efficient ML model for lung cancer cell detection. We aimed to prevent overfitting, reduce complexity, and improve interpretability without sacrificing the model’s accuracy.

## 2. Methodology

### 2.1. Data Collection and Preprocessing

The dataset used for this study consisted of whole-slide images (WSIs) collected from lung tissue biopsies of patients in the Pulmonary Department at the Greater Poland Center of Pulmonology and Thoracic Surgery. All the slides were collected from adenocarcinoma patients, and were of a mixed type (micropapillary, solid, and acinar) but with different percentages of each. We obtained 170 WSIs stained with Hematoxylin and Eosin dye (H&E). Each slide was used to randomly extract an average of six images by zooming in and out in different slide regions and rotating the slide. This process resulted in a total of 934 images: 557 cancerous images and 377 healthy images. The images were further split into training and test datasets. We followed the known approach of a 70% training dataset versus 30% testing dataset [6] as below:The training dataset (71% of total images, n = 662 images) comprised 401 cancerous and 261 healthy images.This was split further to training and validation datasets with 90% (n = 596 images) and 10% (n = 66 images). The training dataset was used to train the model’s parameters. In contrast, the validation dataset was used to fine-tune the model and optimize hyperparameters. We followed National Cancer Institute criteria to differentiate between normal and cancer cells [7].The test dataset (29% of total images, n = 272) included 156 cancerous and 116 healthy images. It was employed to assess the model’s final performance and generalization capability.

These slides were generated using the Ventana Software 3.2v (Ventana Medical Systems, Inc., Oro Valley, AZ, USA), which produces full-color images with Red, Green, and Blue (RGB) channels, while the algorithm code was written using Python v3.12.2 Tensorflow. To ensure consistency and computational efficiency, the images were resized to a uniform height and width of 256 pixels while maintaining the three RGB channels. This resizing process enables the model to learn effectively from the images and provides computational efficiency by reducing the overall data size.

### 2.2. Model Selection

We selected a CNN to operate our algorithm. CNNs fall under the category of deep learning algorithms [8]. Deep learning is a subset of machine learning involving multiple layers of neural networks. The CNN structure forms the backbone of the proposed methodology. CNNs are specifically designed for image analysis tasks, leveraging the inherent spatial structure of images. The primary algorithm used within CNNs is the convolution operation, which involves convolving learnable filters or kernels with the input image to extract relevant features [8].

In the context of lung cancer detection, the convolutional layers of the CNN architecture play a vital role. Each convolutional layer consists of multiple filters that scan the input biopsy images and capture features such as edges, textures, and patterns. These features are learned through the iterative optimization process during model training. By stacking multiple convolutional layers, the network can capture increasingly complex and abstract features, facilitating the discrimination between cancerous and non-cancerous regions in the images.

### 2.3. Separable Convolutional Neural Networks

The architecture incorporates Separable Convolutional Neural Networks (SepCNNs), which enhance feature extraction while reducing computational complexity. SepCNNs decompose the standard convolution operation into two distinct functions: depthwise convolutions and pointwise convolutions. Depthwise convolutions apply a single filter per input channel, independently processing the spatial information within each channel. This operation captures spatial features and enables the network to understand the local structure of the biopsy images. On the other hand, pointwise convolutions employ 1 × 1 convolutions to mix the information from different channels. This operation captures channel-wise features and allows the network to model interactions between various components. SepCNNs significantly reduce the number of parameters and computations required compared to traditional convolutional layers by separating spatial and channel-wise operations. This reduction in complexity enhances computational efficiency and helps prevent overfitting by reducing the risk of model capacity surpassing the available data.

### 2.4. Residual Blocks

Residual blocks play a crucial role in facilitating the training of deeper neural networks, addressing the challenges of vanishing gradients, and promoting the extraction of intricate features. It consists of multiple layers, typically including batch normalization, separable convolution, and skip connections. The batch normalization normalizes the activations of the previous layer, ensuring stable and consistent input distributions throughout the network. This normalization helps improve the gradient flow during backpropagation, enabling more efficient training and reducing the likelihood of getting stuck in suboptimal solutions. The separable convolutional layers and the residual block maintain the improvement process of feature extraction. They employ depthwise and pointwise convolutions as described earlier, capturing spatial and channel-wise features separately. This separation allows for a better representation of complex patterns in the biopsy images. Skip connections or identity mappings form the distinctive characteristic of residual blocks. These connections will enable the gradient to flow directly from earlier to subsequent layers. By introducing these shortcuts, residual blocks enable the network to learn residual mappings, focusing on the difference between the input and output of the block. This mechanism alleviates the vanishing gradient problem, allowing the network to propagate gradients effectively during training. As a result, deeper networks can be trained more efficiently and effectively, promoting the extraction of hierarchical features crucial for accurate cancer detection. The residual blocks are often repeated multiple times in the architecture to allow the network to capture increasingly complex features. Each repetition provides further refinement and abstraction of the learned representations, enhancing the model’s discriminative power.

### 2.5. Model Architecture

The architecture begins with rescaling the input images and dividing the pixel values by 255 to normalize them within the range of 0–1 (Figure 1 and Appendix A). Normalization facilitates more effective learning by bringing the input data into a consistent scale and range. The subsequent convolutional layers employ learnable filters to extract features from the input images. These convolutional layers detect various patterns, edges, and textures, enabling the model to understand the visual characteristics of the biopsy images. Then comes the core component of the architecture, which is the residual block. Following the residual blocks, a Global Average Pooling layer is applied to reduce the spatial dimensions of the feature maps. This pooling operation aggregates the spatial information and extracts the most relevant features while significantly reducing the number of parameters. This process improves computational efficiency and helps prevent overfitting by reducing the complexity of the model. To further prevent overfitting and promote generalization, a dropout layer was included. This layer randomly sets a fraction of the input units to zero during training, reducing the model’s reliance on specific features and enhancing its ability to generalize to unseen data. The final layer of the architecture is a one-neuron layer with a sigmoid activation function. This layer predicts the likelihood of the biopsy image containing cancerous cells, providing a binary classification decision.

### 2.6. Model Optimization and Training

To optimize the model, we used Adam optimizer. Adam is an adaptive learning rate optimization algorithm that combines the advantages of both AdaGrad and RMSProp optimizers [9]. It dynamically adjusts the learning rate based on the gradients of the parameters, allowing for more efficient and effective updates during training. The learning rate started at 0.01, a relatively higher value enabling more extensive initial parameter updates. However, a learning rate decay scheduler was implemented to ensure fine-tuning and convergence. This scheduler reduced the learning rate by 0.25 every 10 epochs. The step decay approach ensured that the learning rate decreased gradually, allowing the model to make more precise updates and converge to an optimal solution. The training process involved iterating over the training dataset for 100 epochs. An epoch represents a complete pass through the entire training dataset. During each epoch, the model adjusted its parameters based on the training and validation data, gradually improving its performance.

### 2.7. Statistical Analysis

PQStat Software (2021, v.1.8.2) was used for statistical analysis. Pearson correlation was used to correlate the years of experience with pathologist performance, and data were reported as r Pearson with a 95% confidence interval (CI). A *p*-value < 0.05 was considered statistically significant.

## 3. Results

### 3.1. Accuracy and Validation

We evaluated the model using accuracy and Area Under the Curve (AUC) metrics on the training, validation, and test datasets (Figure 2). Accuracy represents the percentage of correctly classified samples out of the total number of samples. Achieving high accuracy across all datasets indicates the model’s ability to learn and generalize well. Meanwhile, AUC is a commonly used metric in machine learning and statistics, particularly in binary classification problems. AUC is a measure of the performance of a classification model, and expresses the ability of the model to distinguish between positive and negative classes across various threshold values for the predicted probabilities. The reported results are shown in Table 1.

Moreover, the code was validated on external datasets (n = 105 images) from an open-source repository: the Department of Pathology and Laboratory Medicine at Dartmouth–Hitchcock Medical Center (DHMC) [10]. Remarkably, our model achieved an impressive 100% accuracy, confirming the original results from Dartmouth–Hitchcock (Appendix A).

### 3.2. Model Application

Accuracy and AUC results demonstrate effectiveness of the model in accurately detecting lung cancer from biopsy images. Furthermore, the decision-making process was based explicitly on the cancerous cells found within the images. This observation was confirmed by analyzing the last convolutional layer’s output, highlighting the model’s capability to distinguish cancerous cells from normal cells with high sensitivity, showing the area where the model recognizes the cancer cells (Figure 3). It is worth emphasizing that the model can detect and interpret the input image within less than 10 s, giving highly accurate and precise results.

### 3.3. Comparison of the Model Results with the Pathologist’s Decision

Ten active pathologists from the Pulmonary Department at the Teaching Hospital of Poznan University of Medical Sciences participated in the study (average years of experience 11.9 ± standard deviation 10.13). They were each provided with exactly fifteen images and were asked to classify each slide as healthy, cancerous, or uncertain (Appendix A). While the model could accurately classify all the slides, the pathologists’ accuracy ranged from 0 to 100%, with an average accuracy of 79%. Moreover, the model’s overall performance surpassed that of active pathologists, with an accuracy of 100%.

It is worth mentioning that all pathologists correctly identified the slides with healthy cells. However, their accuracy declined when they were uncertain about some slides, particularly slide number 7, which all pathologists were uncertain about. We observed a significant linear correlation between pathologists’ accuracy and years of experience (r Pearson = 0.71, 95% CI 0.14 to 0.93, *p* = 0.022) and the linear regression equation of accuracy (y = 72.61 + 0.62x, as x denoted for years of experience), as shown in Table 2.

## 4. Discussion

### 4.1. Main Findings

The convolutional layers in the model improved cancer cell detection, and incorporating SepCNNs reduced computational complexity and overfitting risks, while residual blocks and batch normalization facilitated the training of deeper spots by addressing challenges like vanishing gradients and ensuring stable input distributions. These designed and integrated elements collectively improve the model’s accuracy and efficiency in distinguishing between tumor and healthy cells, with an accuracy between 96% and 98%. Our model was designed to handle complex nonlinear relationships, fault tolerance, and parallel distributed processing, and pay attention to the true positives. Thus, it can capture and detect fine-tuned details in the training slides and then generalize the outcomes in a new dataset.

The model was successfully deployed in clinical practice with histopathological slides, and its results were compared with those of active pathologists. The implementation of such an AI tool in clinical settings is highly pertinent in the context of lung cancer, particularly in Poland.

### 4.2. Comparison with Existing ML Models in Lung Cancer Detection

Many successful models were created to distinguish between healthy and tumor lung cells using the CNN approach; for instance, Gürsoy et al. [11] developed an AI-based model employing a CNN for comparative diagnostic evaluation against human pathologists. The dataset comprised 158 nodules extracted from lung cancer patients, with a distribution of 77 malignant and 81 benign cases, assessed independently by two radiologists and pathologists. The diagnostic outcomes exhibited a striking concordance, revealing an accuracy of nearly 91% between the AI model and human assessments.

Wu et al. [12] undertook a project to delineate the advantages conferred by integrating a DL-based model to enhance diagnostic precision and efficiency within pathology. Employing the CNN system, their focus centered on identifying non-small cell lung cancer tumors, aiming to augment the diagnostic capabilities of pathologists. The study encompassed a dataset of 173 WSIs meticulously evaluated by human pathologists, partitioning 70% for training data and 30% for testing and validation. The reported findings revealed 93% accuracy and 96% specificity. The application of AI-assisted diagnostic tests further affirmed the potential enhancement in repeatability and efficiency for untrained pathologists utilizing the AI model. Also, Xie et al. [13] introduced an innovative interdisciplinary approach in the context of lung cancer diagnosis in China. The study entailed the utilization of six ML algorithms, including K-nearest neighbor, Naïve Bayes, AdaBoost, Support Vector Machine, Random Forest, and a neural network, employing a 10-fold cross-validation technique for the early detection of lung cancer. A dataset comprising WSIs from 110 lung cancer patients and 43 healthy participants was analyzed. The outcomes revealed a notable accuracy of 98.9% and a sensitivity of 98.1%. However, the model still could not integrate the plasma metabolic biomarkers with computed tomography screening or other lung tumor features, which could improve the model’s performance.

Moreover, Dritsas et al. [14] applied the ML-based model “Rotation Forest” to detect the early stage of lung cancer. The WSI dataset was collected from 309 participants, and the model’s evaluation was grounded in precision signifying the measure of quality, recall representing the measure of quantity, and F-Measure, which embodies the harmonic mean of precision and recall, facilitating a comprehensive evaluation through a singular metric. The reported outcomes indicated an AUC of 99.3% and F-measure, precision, recall, and accuracy metrics at 97.1%. However, this ML model was based on a publicly available dataset, and it did not come from a hospital unit or institute, which affects the features input and characters in addition to the interpretability and outcome of the model.

### 4.3. Clinical Implications

Lung cancer is prevalent and poses a significant healthcare challenge, requiring efficient diagnostic tools to cope with the escalating workload on pathologists [15]. The surge in lung cancer cases, coupled with the evolving intricacies of histopathological evaluations following guideline modifications, contributes to an augmented burden on pathologists. Unfortunately, there is a simultaneous diminution in the pathology workforce in Poland. Adjusted for the increasing number of new cancer cases annually, the workload per pathologist in Poland has increased, signifying a growing disparity that may lead to delayed cancer diagnoses and diagnostic inaccuracies. Based on the latest published articles, Poland has less than 800 active pathologists, indicating one pathologist per 51,000 citizens, one of the lowest in the European countries, as the average is one pathologist per 40,000 citizens [16]. The statistics mentioned that around 30,000 deaths per year are due to lung cancer. One of the factors that could have a vital role here is the prolonged diagnosis due to a shortage of active pathologists, the time-consuming process, and ineffective treatment. To the best of our knowledge, the implementation of AI applications in medical diagnosis inside Polish hospitals is still inadequate, and designing a dedicated model considering the pathologists requirements gathered earlier will definitely be helpful in the decision-making process. It can improve the accuracy and efficiency of medical diagnosis and support pathologists in their daily tasks [4]. To the best of our knowledge, this is the first study in Poland to incorporate a DL model design with pathologists, considering their needs and involving them in the development phase.

Eventually, the model could be used by universities to train and educate medical students for a better understanding of the detection process and highlight the advantages of using AI applications in the medical field.

### 4.4. Model Limitations

Even though our model provided a high precision rate exceeding 96% and detection power within 10 s, pinpointing the areas where tumor cells were detected, several limitations need to be addressed. First, acquiring large datasets for training, testing, and validating the ML-based model is challenging; the more significant the dataset, the higher the accuracy and sensitivity of the model. Our study collected 170 patient slides, resulting in 934 images for our model. Our model will develop a more nuanced understanding of tumor properties with larger datasets, which will be reflected in the final output and interpretation. Finally, the lack of a standardized lung tumor pathology image database hinders the diagnostic efficacy of ML models, underscoring the need for continuous improvements in ML technology to ensure robustness and consistent recognition across diverse databases from various regions.

### 4.5. Ethical Considerations and Potential Social Impacts

When deploying AI algorithms for cancer diagnosis, several ethical considerations and potential social impacts should be carefully addressed. First, clinicians must consider patient autonomy, as they may have different attitudes toward risk and preferences regarding false-positive and false-negative results [17]. Still, it is essential to mention that the transformation into an AI model is a complex process requiring in-depth validation and constant supervision and maintenance to ensure its efficacy in clinical settings [18]. Second, Kiseleva et al. [19] recommended that AI’s transparency in healthcare be viewed as a multilayered system of measures. The three layers of transparency to be considered when using AI applications in healthcare are external (from physicians toward patients), internal (from AI providers toward physicians), and insider (from AI providers toward themselves) [19]. Third, several data privacy and security issues were reported, which require proper legalization, such as the General Data Protection Regulation in the European Union [20]. It is worth mentioning that some manufacturers of such AI tools can also provide their solution to run mainly on local servers without the need to connect to the internet, hence providing the healthcare facility more control over their data. Fourth, some concerns have been raised previously on AI replacing physicians and its influence on the patient–physician relationship [21]

## 5. Conclusions

We developed a model based on optimized residual blocks to enhance feature representation without compromising detection accuracy (96% to 98%) within a swift detection time (below 10 s). The model outperformed results reported by active pathologists, suggesting that an AI algorithm assisted by pathologists can enhance diagnostic skills and reduce the lead time in the diagnosis process. Universities and healthcare facilities may utilize the code to train medical students and junior physicians due to the fact that accuracy of pathologists has improved over their years of experience. Future AI algorithms could be developed to detect different stages of cancer, and further research on the current algorithm’s scalability and deployment could be conducted.

## Figures and Tables

**Figure 1 cancers-16-01506-f001:**
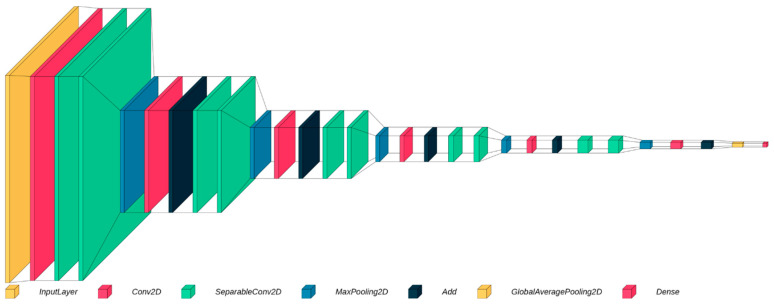
A summary of the model legend: (i) Input preprocessing [inputLayer]: rescales input images and normalize pixel values to the range of 0–1. Effective learning benefits from consistent scaling. (ii) Convolutional layers [Conv2D and SeparableConv2D]: learnable filters extract features (patterns, edges, and textures) from input images and enable the understanding of visual characteristics in biopsy images. (iii) Residual blocks [sequence of 5 layers/operations: 1x MaxPooling2D, 1x Conv2D, 1X Add (operation), and 2X SeparableConv2D]: core component of the architecture that enhances feature representation. (iv) Global Average Pooling [GlobalAveragePooling2D]: reduces spatial dimensions of feature maps, extracts relevant features while minimizing parameters, improves computational efficiency, and prevents overfitting. (v) Dropout layer: randomly sets input units to zero during training, enhancing generalization by reducing reliance on specific features. (vi) Final layer [Dense]: one-neuron layer with sigmoid activation predicts the likelihood of cancerous cells in biopsy images (binary classification).

**Figure 2 cancers-16-01506-f002:**
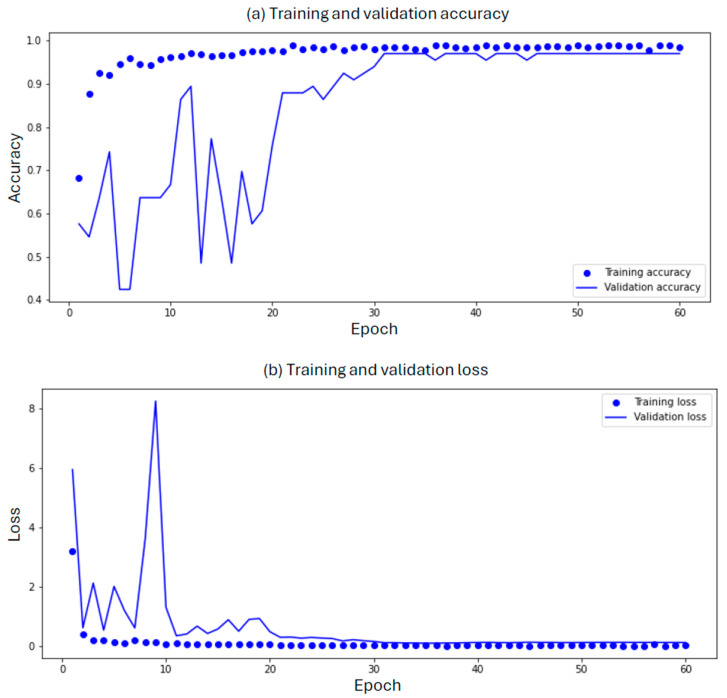
(**a**) Training and validation accuracy: The blue line represents the training accuracy over multiple epochs, indicating how well the model performed on the training data. The dotted circles correspond to the validation accuracy, reflecting the model’s generalization ability on a separate validation dataset. The x-axis represents the number of training epochs, while the y-axis shows the accuracy, ranging from 0 to 1. (**b**) Training and validation losses: In the second subplot, the blue line represents the training loss (error) as the model learned. Lower training loss values indicate better convergence. The blue dotted circle curve represents the validation loss, quantifying the discrepancy between predicted and actual values during validation. Similar to the first subplot, the x-axis corresponds to the number of training epochs, and the y-axis represents the loss.

**Figure 3 cancers-16-01506-f003:**
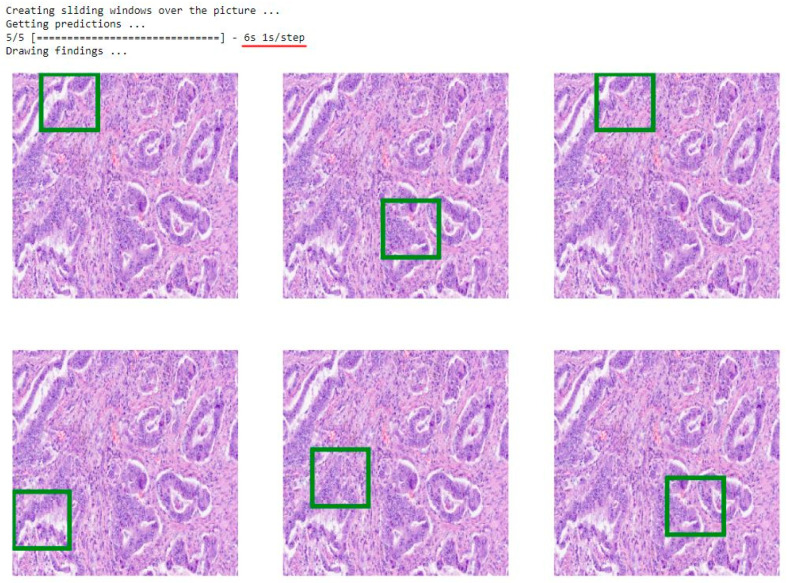
This figure presents a screenshot of the model’s demo results. The text above the screenshot provides immediate feedback after initiating the command. Equal signs indicate loading progress, and the total time taken by the model to generate the figures is displayed at the end. In this example, the model completed the prediction task in 6 s (averaging 1 s per step). The model then automatically generated six identical images based on the input. These images corresponded to the same original image that was inserted. Additionally, the model identified the most optimal regions of interest where cancer cells had been detected. Green rectangles highlight these regions. In the example above, there are six distinct spots, each indicating a region where the presence of cancer was determined.

**Table 1 cancers-16-01506-t001:** Accuracy and AUC for the training, validation, and testing datasets.

Dataset	Accuracy	AUC
Training dataset	98%	99%
Validation dataset	96%	97%
Testing dataset	97%	98%

**Table 2 cancers-16-01506-t002:** Comparisons between model and pathologist decisions on 15 exact images (H, healthy; C, cancer; UC, uncertain).

Image nr.	Model			Pathologist Decision (n = 10)										Pathologist Slide Accuracy (%)
	Decision	Array	Accuracy(%)	P1	P2	P3	P4	P5	P6	P7	P8	P9	P10	
1	H	<0.001	100	H										100
2		<0.001	100											100
3		<0.001	100											100
4		0.05	100											100
5		0.004	100											100
6		<0.001	100	H									C	90
7	C	1	100	UC										0
8		1	100	UC		C	UC		C	UC				20
9		1	100	C										100
10		1	100											100
11		1	100	C									UC	90
12		1	100	UC	C									70
13		1	100	C	UC	C	UC		C	UC		C		40
14		1	100	C										100
15		1	100	UC		C								80
Pathologist year of experience				11	11	14	12	12	14	12	11	13	10	
Individual pathologist accuracy (%)				73.33	73.33	93.33	80.00	80.00	93.33	80.00	73.33	86.67	66.67	

## Data Availability

Data are contained within the article and Appendix A.

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
