# Peer review of "Development and Validation of a Deep Learning Model for Histopathological Slide Analysis in Lung Cancer Diagnosis"

_cancers, 2024, doi:10.3390/cancers16081506_

Round 1

Reviewer 1 Report

Comments and Suggestions for Authors

Comments on the Quality of English Language

Author Response

Please find our attached response

Reviewer 2 Report

Comments and Suggestions for Authors

Major points.

The article deals with utmost important issue of implementing AI into the process of cancer diagnosis. Multiple groups have demonstrated a multimodal approach in the transformer approach, which harnesses the capabilities of Foundation models.

Nevertheless this manuscript undoubtedly requires some revisions in order to make more clear and informative:

1. lines 2-3: the title is somehow misleading in suggestion that there is "diagnosis of HP slide" Plain reformulation will give the title the proper meaning

2. line 33: the statement on sparing time to active pathologists should be either expanded or omitted. In this form it seems to be too overreaching. Additionally in the abstract it have to be stated more categorically whether the model "will enhance" or enhances diagnostic procedure.

3. line 58: "cancer cell diagnose" - reformulate please

4. The methods are not obvious to this reviewer. you use 170 WSI from which you randomly generate 934 slides from biopsy specimen. How do you secure randomness and how it is feasible to acquire from small specimen many slides, not mentioning that it does confine variability.

5. line 67: it is not appropriate notation if you mean 934 images is a sum of 557 and 377.

6. FIGURES: the figures composition is unacceptable. In all the them the legend is missing. In Fig. 1 the abbreviations have been not previously mentioned in text. Fig. 2 and Fig. 4 are totally unreadable. Fig 4. is the mixture of lines of code or prompts (?) followed by HE-stained slides with rectangles placed on a random way with no explanation at all. Generally speaking the results presentation is suboptimal.

7. DISCUSSION: one-pager comprising three articles.

8. CONCLUSION: no conclusion provided. The text is a mixture of discussion , comments and general view,

Comments on the Quality of English Language

moderate/extensive editing needed

Author Response

Please find our attached response

Round 2

Reviewer 2 Report

Comments and Suggestions for Authors

The Authors have considerably improved the readability of the article, especially adding figure captions.

However there are still some minor points to be corrected.

1. Fig. 1 - Beginning of caption should be corrected. Numbering i through vi is missing in the figure

2. Still 4 lines of code (or whatever it is) above Fig. 2 looks a bit messy

3. Discussion has been divided to subparagraphs, which improved clarity. One section has been added. Yet only three references are discussed. When putting  the string: "machine learning AND pathological specimen AND lung cancer" it returns 30 citations.

4. Unfortunately still the conclusion section resembles an abstract or the Authors reflections on the subject.

Comments on the Quality of English Language

As stated above